# Cancer Stem Cells from Definition to Detection and Targeted Drugs

**DOI:** 10.3390/ijms25073903

**Published:** 2024-03-31

**Authors:** Barbara Ruszkowska-Ciastek, Katarzyna Kwiatkowska, Dorinda Marques-da-Silva, Ricardo Lagoa

**Affiliations:** 1Department of Pathophysiology, Faculty of Pharmacy, Nicolaus Copernicus University, Collegium Medicum, 85-094 Bydgoszcz, Poland; 2Department of Laboratory Diagnostics, Jan Biziel University Hospital No. 2, 85-168 Bydgoszcz, Poland; katarzynakwiatkowska@abs.umk.pl; 3Laboratory of Separation and Reaction Engineering-Laboratory of Catalysis and Materials (LSRE-LCM), Polytechnic Institute of Leiria, 2411-901 Leiria, Portugal; dorinda.silva@ipleiria.pt (D.M.-d.-S.); ricardo.lagoa@ipleiria.pt (R.L.); 4Associate Laboratory in Chemical Engineering (ALiCE), Faculty of Engineering, University of Porto, Rua Dr. Roberto Frias, 4200-465 Porto, Portugal; 5School of Technology and Management, Polytechnic Institute of Leiria, Morro do Lena-Alto do Vieiro, 2411-901 Leiria, Portugal

**Keywords:** cancer stem cells, leukaemia stem cells, cancer progression, solid tumours, haematological malignancies, tumour microenvironment, artificial intelligence, targeted therapy, polyphenols

## Abstract

Cancers remain the second leading cause of mortality in the world. Preclinical and clinical studies point an important role of cancer/leukaemia stem cells (CSCs/LSCs) in the colonisation at secondary organ sites upon metastatic spreading, although the precise mechanisms for specific actions are still not fully understood. Reviewing the present knowledge on the crucial role of CSCs/LSCs, their plasticity, and population heterogeneity in treatment failures in cancer patients is timely. Standard chemotherapy, which acts mainly on rapidly dividing cells, is unable to adequately affect CSCs with a low proliferation rate. One of the proposed mechanisms of CSC resistance to anticancer agents is the fact that these cells can easily shift between different phases of the cell cycle in response to typical cell stimuli induced by anticancer drugs. In this work, we reviewed the recent studies on CSC/LSC alterations associated with disease recurrence, and we systematised the functional assays, markers, and novel methods for CSCs screening. This review emphasises CSCs’ involvement in cancer progression and metastasis, as well as CSC/LSC targeting by synthetic and natural compounds aiming at their elimination or modulation of stemness properties.

## 1. Introduction

Despite extensive research into the nature of cancers, they remain a leading cause of death. Based on the GLOBOCAN 2020 registry with over 10 million fatal events in 2020 alone, including lung cancer (1,796,144), colorectal (935,173), liver (830,180), stomach (768,793), breast (684,996), oesophagus (544,076), pancreas (466,003), prostate (375,304), cervix uteri (341,831), leukaemia (311,594), and so on [1]. Thus, many questions are still to be answered in this regard: What else do we need to know to tame cancers? Have we reached the glass ceiling in this regard? How long is this path? Or perhaps, is this exploration a never ending story?

Many authors have defined cancerous process and its sounds like a mantra. The phenomenon has been explained by accumulation of genetic and epigenetic alterations, which enhance cell transformation into a specific (cancerous) phenotype, i.e., limited apoptosis, infinite replicative capacity, increased motility, and pro-angiogenic ability [2,3]. It is also important to mention the altered energetic metabolism (the Warburg effect) and facility to convert into endothelial-like cells in order to maintain metabolic balance in tumour-dependent hypoxic areas (vascular mimicry), and the potential ability to enter and exit a quiescent state and immune evasion by cancer cells, which are also fundamental features of cancer transformation [4,5]. Delving deeper into the topic, we need to account for the role of cancer stem cells (CSCs), which constitute a specific tumour cell population. These cells show particular characteristics like localisation within the tumour, promotion tumour initiation, a high capacity to create colonies, pro-metastatic, pro-recurrence, and, last but not least, low drug sensitivity. According to the literature, CSCs exhibit great pro-neoplastic potential by maintaining pre-neoplastic foci, i.e., ideal tumour-initiating environments [2,6,7]. Furthermore, CSCs possess the potential to differentiate into multiple cell lineages, including pericytes, endothelial cells, or cancer-associated fibroblasts, and they are able to remodel their microenvironment, which enables the recruitment of other cells; consequently, they participate in the tumour growth and spreading [7,8].

In spite of great advances, modern chemotherapeutic agents and immunotherapies have not eliminated the severe worldwide cancer mortality [9,10]. Standard anticancer agents do not distinguish normal cells from cancer cells; thus, the chemo-related side effects are common and cause serious discomfort among oncology patients [11]. The individual/intrinsic profile of each patient must be taken into consideration, since therapeutic failures might be associated with de novo lower sensitivity to drugs or the acquisition of treatment resistance as a result of the therapy used [12]. The insufficient treatment response may be due to the heterogeneity of cancers, which is also associated with CSC biology [2,13]. Thus, the main purpose of stratified medicine is to translate the molecular status of tumour cells into predictive and prognostic indexes that can be applied to personalise treatments leading to longer survival and reduced toxicity [10,14]. In this line, patients who are stratified as high risk for relapse could be treated with adjuvant mode, while patients without detectable CSCs after neoadjuvant treatment and surgery might be adequate for less intensive follow-up procedures. Well-defined risk factors that are related to shorter survival rate in oncology patients include advanced age, unfavourable genetic profile, associated comorbidities, as well as overtreatment and treatment-related toxicity. The ideal balance between a patient’s risk and favourable outcomes are of utmost relevance to providing a therapy decision [9,10,11].

In this review, we discuss the general concept, characteristics, and detection technologies of CSCs and leukaemia stem cells (LSCs). Further, we highlight recent advances in the development of drug candidates targeting CSCs/LSCs.

## 2. General Concept of Cancer Stem Cells

Already in 1838, Johannes Müller, and subsequently in 1858, Rudolf Virchow, suggested the hypothesis of the embryonic origin of tumour cells, which was confirmed by Julius Cohnheim in 1877 and further studies (Figure 1) [5,9,15,16]. CSCs were identified for the first time in an acute myeloid leukaemia (AML) model, and to this day, presenting in virtually all cancer types, by employing various cluster differentiation (CD) markers or through side population examination [6,9,10,17,18,19].

The discovery of CSCs/LSCs has modified the understanding of cancer’s nature and its response to anticancer drugs. Nowadays, it is believed that CSCs are responsible for the formation and expansion of cancerous tissue. CSCs, also called tumour-initiating cells [4,10] or stemness-high cancer cells [10], exhibit major stem-like properties, including self-renewal ability, pluripotent potential, and clonogenicity, which may promote the establishment of a metastatic foundation and resistance to standard chemotherapies and radiation [6,9,10]. Interestingly, Kreso and Dick demonstrated that only particular and more aggressive CSCs show the potential for tumour expansion and relapse, so various populations or even a hierarchy of CSCs may be present in the tumour bulk [20].

**Figure 1 ijms-25-03903-f001:**
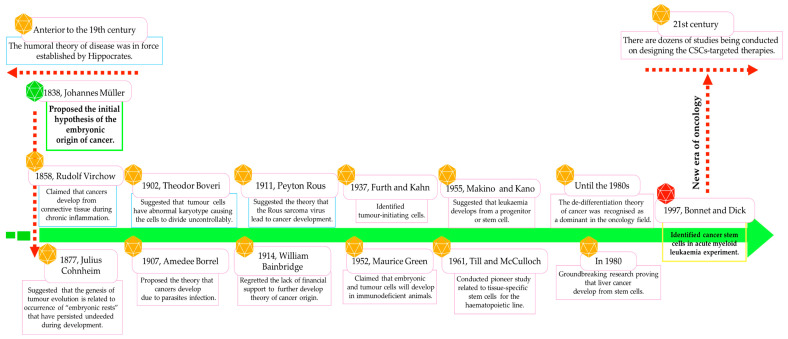
Most representative milestones in the advancement of knowledge about the nature of cancer/leukaemia stem cells.

### 2.1. Cancer Stem Cells: Origin and Detailed Characteristics

Stem cells rarely divide under physiological conditions. In bone marrow, only 10% of stem cells are in the replication, pre-division, or mitosis phases at the same time, which proves that, despite unlimited proliferative activity, stem cells divide relatively rarely [3,21]. It is believed that CSCs may develop from normal stem cells or partially differentiated progenitor cells present in a given niche, or they may originate from fully differentiated somatic cells, so CSCs manifest a similar phenotype as normal stem cells (Table 1). However, the transformation of a normal stem cell into a CSC requires that a few conditions be met, including loss of cell cycle control and the accumulation of genetic and epigenetic alterations [2,17]. Toh et al. pointed out that epigenetic alterations (i.e., DNA methylation, chromatin remodelling, and histone modifications) are among the first events promoting the transition of stem cells into CSCs. Production of CSCs is also due to a decline in the expression of tumour suppressor genes, especially TP53, ATM, PTEN, and others [17].

CSCs are difficult to eradicate—they overexpress drug efflux pumps, secrete detoxifying enzymes, and demonstrate a potent ability to stimulate anti-apoptotic and pro-survival pathways, as well as DNA repair. The currently used chemotherapy, which acts primarily on rapidly dividing cells, is unable to adequately affect CSCs with a low replication index [4,6,18]. One of the proposed mechanisms of CSC resistance to anticancer agents is the fact that these cell populations can easily manoeuvre between different phases of the cell cycle in response to typical cell stimuli induced by anticancer drugs. Thus, CSCs in the G0 phase are insensitive to cell cycle blockade signals followed by failure of the apoptotic cascade, which gives them the potential to survive longer in a dormant state [22]. To understand this relationship between cancer progression and CSCs, we should first recognise the nature of CSCs. Thus, the most relevant differentiating and characterising features of normal stem cells and CSCs are included in Table 1 [2,3,4,6,9,10,13,23,24,25,26,27,28,29,30].

**Table 1 ijms-25-03903-t001:** Features of normal stem cells versus cancer stem cells (CSCs).

Features	Normal Stem Cells	Cancer/Leukaemia Stem Cells
Localisation	In almost all physiological tissues	Periphery of the tumour
Composition	Hierarchical structure	Hierarchical structure
Characteristics	Primitive or undifferentiated precursors	Initiate and reconstitute tumour lesions
Function	To maintain tissue homeostasis	To maintain the unlimited growth of tumours and their morphological diversity
Self-renewal	Potent	Potent (tumour re-creation by metastasis)
Differentiation pattern	Pluripotent (differentiate into different kinds of normal cells)	Pluripotent (differentiate into different kinds of cancer cells)
Cell differentiation	Balanced	Dysregulated
Cell division	Mostly asymmetric	Subpopulations of CSCs: * Early-stage CSCs—mostly asymmetric * Late-stage CSCs—mostly symmetric
Cell cycle phase	G0/G1 phase	Ability to switch into any phase (mostly slow-cycling behaviour)
Proliferation index	Low, unlimited and well-controlled	Varied and uncontrolled
Morphology characteristics	High nuclear-to-cytoplasmic ratios	High nuclear-to-cytoplasmic ratios
Migration ability	High	High
Cell phenotypic potential (cell plasticity)	Stable	Heterogeneous
Partner of sld five 1 detection	Negative	Positive
Pro-angiogenic property	Limited	Unlimited
Drug sensitivity	Moderate	Strong resistance
Selected surface markers	CD24^+^, CD34^+^, CD44^+^, CD90^+^, CD133^+^	CD24^−/low^, CD34^+^, CD44^+^, CD90^+^, CD133^+^, ALDH1^high^, ESA, EpCAM, side population cells
Immunosuppressive effect	Negative	Positive (via paracrine manner)
Survival rate	Prolong	Enhances their survival in an autocrine manner
Apoptosis	Antiapoptotic phenotype	Antiapoptotic phenotype (mediated by IL-4)
Chromosomal abnormality	Normal karyotype	Subpopulations of CSCs: * Early-stage CSCs—normal karyotype * Late-stage CSCs—an abnormal chromosome number
Telomerase activity	Potent	Potent
Histone H3 demethylation	Positive	Positive
Expression of Oct4, Notch, Sox1 genes	Positive	Positive
DNA repair ability	Potent	Potent
Genetic stability	Normal	Lost
Presence in peripheral blood	Trace amounts	Trace amounts
% of cells in specific tissue	0.01	0.02–25

Notch, Oct4, Sox1: specific genes for all stem cells; ALDH1: aldehyde dehydrogenase 1; CD24: a small surface protein responsible for cell–extracellular matrix (ECM) and cell–cell interactions; CD34: a transmembrane glycoprotein expressed on early lymphohematopoietic stem cells, progenitor cells, and endothelial cells; CD44: a multifunctional glycoprotein responsible for cell adhesion, signalling, proliferation, migration, haematopoiesis, and lymphocyte activation; CD90: a glycophosphatidylinositol (GPI) anchored conserved cell surface protein; CD133: also known as prominin-1, a transmembrane cell surface glycoprotein commonly utilised as a hematopoietic stem cell marker; CSCs: cancer stem cells; DNA: deoxyribonucleic acid; EpCAM: epithelial cell adhesion molecule; ESA: epithelial-specific antigen; IL-4: interleukin 4; * It means that Early-stage CSCs and Late-stage CSCs are subpopulations of CSCs.

### 2.2. The Importance of the Tissue-Specific Microenvironment for the Maintenance of CSCs/LSCs

A niche as a specific microenvironment ensures suitable conditions for stem cell development and maintenance. The stem cell niche refers to the space in which stem cells are kept ready for the self-renewal, cell division, and differentiation necessary to maintain tissue homeostasis [2,27,31]. The specific features of niches for CSCs are disruption of the immune system and accumulation of malignant cells [2,24,29,32]. In this context, it is important to take into account that chronic inflammation is a natural driver in cancer-triggering niches. Specific characteristics of CSC niches are maintained by accumulation of cancer-associated fibroblasts, tumour-associated macrophages, tumour-associated neutrophils, and cell-mediated adhesion, which regulate cell–cell interactions and stromal, endothelial, and T cells [2,18,29,33,34]. Additional elements including extracellular vesicles, soluble factors, and the extracellular matrix support cancer-related surroundings [18]. Such a microenvironment favours specific features of CSCs, including infiltration, metastasis, and stimulation of tumour-associated neovasculature [18,35]. It is well known that neoangiogenesis is triggered in low-oxygen regions, but the neovasculature network is abnormal due higher permeability and a twisted, immature structure [36]. Additionally, hypoxic niches maintain undifferentiated CSCs via limiting cell cycling, followed by cell division rate reduction (stimulates switch into G0 phase) [24,28]. Interestingly, cancer-dependent hypoxia triggers a protective environment against DNA damage. According to Mohyeldin et al., 20% oxygen saturation was associated with significantly higher tissue damage compared to 3% O_2_ [37]. The above-mentioned mechanisms lead to the formation of pro-metastatic sites and also contribute to the insensitivity of hypoxic niches to chemotherapy [24]. Hypoxia-inducible factors (HIFs) affect the cell division, self-renewal, and cancerogenicity of CSCs. In accordance, higher CD44^+^ and CD133^+^ expression in hypoxic conditions was noted by Bai et al. and Won et al. [38,39].

Disruption of the bone morrow (BM) niche structure is a predictable state in blood malignancies. Accumulation and infiltration of leukaemia cells promotes elimination of normal haematopoietic progenitor cells from the BM niches and prepares an ideal microenvironment for them [40]. This modified BM microenvironment enables typical behaviours of LSCs including self-renewal, dormancy, and apoptosis evasion [41]. Moreover, the modified BM niche remains a space for LSCs, which is a reservoir for residual leukaemia cells and promotion of recurrence [42]. Interestingly, the BM niche demonstrates two separate microenvironmental regions (the osteoblastic niche and vascular niche) that likely modulate the cycling of LSCs [31,43]. Both niches effectively collaborate and promote the self-renewal, cell division, motility, and organisation of BM-related stem cells and LSCs [44].

### 2.3. Immunophenotypic Fingerprints of CSCs/LSCs

CSCs and non-CSCs can be distinguished via specific CD markers, but also based on their self-renewal ability [6]. Nevertheless, the disadvantage of the flow cytometry method is that selected surface markers are co-expressed in both populations. Additionally, in the analysis of the surface markers’ expression patterns, patients’ ethnicity or race must be taken into account in order to standardise the results [45]. Furthermore, considering both solid tumours and haematological malignancies, there is an intra-tumoral heterogeneity of surface markers among one type of cancer and stem cell plasticity, which often produces inconsistent results in this regard (Table 2). Thus, a specific phenotype of CSCs/LSCs is often not yet available in certain cancers. Therefore, this analysis should be extended by including enzymatic analysis (ALDHs) or CSC colony formation ability [45,46]. Table 2 presents composite profiles of CSC/LSC surface markers in solid tumours and haematological malignancies.

### 2.4. Detection of CSCs/LSCs

The current techniques for CSC identification include the estimation of surface markers or its functionality. The expression pattern of surface markers is commonly used for CSC/LSC determination and isolation using fluorescence-activated cell sorting (FACS) [46]. FACS based on detection of CSC/LSC-specific immunophenotype or surface antigens and further segregation of fluorescent vs. non-fluorescent cells can be implemented using a multicomponent assay [46,109,110]. However, determining one specific marker for CSCs or LSCs is very difficult (Table 2), and the method also requires aseptic conditions and vast number of cells. An alternative to FACS is magnetic-activated cell sorting (MACS), which is easy to perform and requires a smaller number of cells. MicroBeads with a typical diameter of 100 nm specifically bind to antigens enabling isolation of the targeted cells without further staining [46,110,111]. However, the selection of cells in mono-parameter mode is the greatest weakness of the method [112]. Undoubtedly, high intra- and inter-tumoral diversity limits the application of a well-defined immunophenotype for effortless detection of CSCs or LSCs, so there is still space for functional methods (Figure 2) [26]. It has been suggested that the assessment of surface markers is not sufficient for the detection of specific pro-metastatic CSCs; hence, a broader view of this issue is required and gene expression profiling of these subpopulations should be included [17]. Thus, the evaluation of CSCs/LSCs requires advanced analytical methods that demonstrate proper sensitivity and specificity as well as limit false positives and false negatives.

In the literature, several in vivo and in vitro functional methods have been proposed to recognise CSCs/LSCs in cancer tissues or cell lines. Figure 2 shows the application, benefits, and weakness of currently used methods for functional assessment of CSCs/LSCs [26,45,46,52,111,113,114,115,116,117]. It is accepted that new diagnostic techniques are indispensable for adequate recognition of these cells. Could the use of artificial intelligence (AI) or deep learning be the answer to the current need to identify CSCs (Figure 3)?

**Figure 2 ijms-25-03903-f002:**
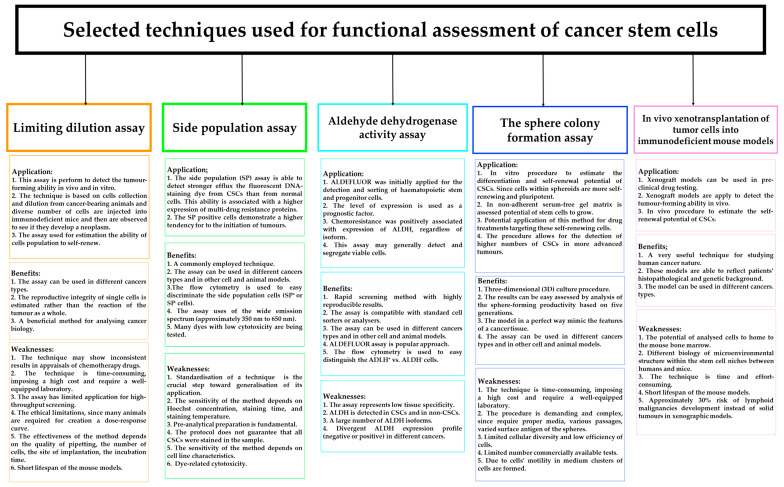
In vivo and in vitro techniques assessing the functionality of cancer stem cells (CSCs) their application, benefits, and weaknesses.

### 2.5. The Space for Artificial Intelligence in Cancer Stem Cell Detection

A new era of cancer diagnostic panels is opening or even forcing the space for AI technology in order to deliver fully automated identification of biological images of heterogeneous stem cell populations, including CSCs [9]. Considering the difficulties in laboratory practice in differentiating between normal and cancer stem cells, AI algorithms can find an important place in CSC detection. However, it is necessary to remember the appropriate and standardised method of selecting CSCs through qualitative and quantitative assessment of its morphological features. Deep learning algorithms are trained, tested, and validated to assess the proliferation, apoptosis, and dormant status of CSCs. The following factors may limit the use of standard AI algorithms: CSCs demonstrate different cell sizes with a different cytoplasmic-to-nuclear ratio in respect to non-CSCs, and the low number of CSCs in cancer tissue. Furthermore, insufficient image contrast and areas with blurry image features pose critical limitations in the training and testing stages of CSC recognition. New technologies open space for faster, automated diagnostics, but algorithms that are not fully developed still have limitations to overcome before being introduced into clinical practice. Figure 3 illustrates the applications of AI in the detection of cancer stem cells [9,23,114,118]. Ensuring proper identification of CSCs by advanced learning models able to include intra- and inter-tumoral heterogeneity will increase the application of AI in this field.

**Figure 3 ijms-25-03903-f003:**
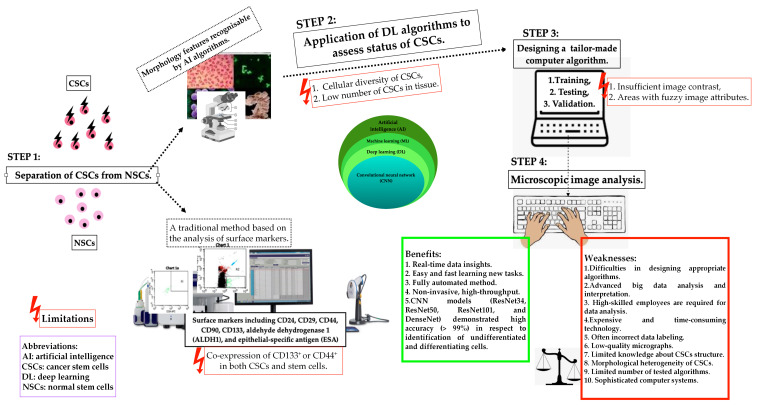
Schematic representation of the applications of artificial intelligence in the detection of cancer stem cells (CSCs), their potential, and associated technical limitations.

## 3. Cancer and Leukaemia Stem Cells in Disease Recurrence

Tumour cell dissemination, from primary origin to secondary sites, is strongly related to cancer-associated mortality in two out of every three solid tumours [119]. The CSC paradigm assumes that solid tumours and leukaemias are hierarchically defined, with CSCs at the top of this pyramid, leading to tumour development, spread, relapse, and drug resistance [6]. Interestingly, higher CSC counts have been detected in leukaemias and lymphomas, while solid tumours presented lower numbers [120]. However, it is considered that higher-grade tumours show higher percentages of CSCs [82,121]. Nevertheless, according to the CSC model, not all of those cells are able to trigger cancer progression (Table 3). Tumour spreading depends on a more anomalous and particular subpopulation of CSCs. Thus, there is a need to identify at least two CSC/LSC subpopulations: early-stage (pre-neoplastic) and late-stage (pro-metastatic) CSCs/LSCs. Table 3 summarises the key characteristics of CSC/LSC subfractions [2,44].

In order to fully understand the relationship between CSCs and cancer progression, it is important to note that dysregulation of vascular homeostasis facilitates tumour progression [35,36]. Transcription factors specific for mesenchymal cells (Twist1, Slug, and Snail) and antigens (Vimentin and N-cadherin) are expressed on the surface of CSCs, helping them to undergo epithelial–mesenchymal transition (EMT) and trigger the formation of secondary malignant phenotypes, cell migration, and apoptosis-resistant CSCs [29,122,123]. Furthermore, the upregulation of stemness-related components, including Oct4, Notch, ALDH1, and SOX1, confirms the ability to effectively switch between CSC and non-CSC states [29].

The EMT is stimulated by mediators released from the niche, i.e., transforming growth factor β (TGF-β), hepatocyte growth factor (HGF), HIF, Hedgehog, Wnt, and Notch [30]. The Wnt/β-catenin, Hedgehog, Notch, and PI3K/Akt/mTOR signalling pathways are upregulated in all solid and non-solid tumours, leading to the enhancement of CSC/LSC-specific properties. The Wnt pathway enhances cancer cell division, motility, and drug resistance, while the self-renewal of CSCs/LSCs is mediated by the Hedgehog and Notch pathways [29,30,52,122,123]. However, the research so far has not allowed for us to fully understand and control the mechanism by which CSCs/LSCs contribute to cancer invasion. Nevertheless, the above-indicated signalling pathways provide a mechanism for explaining the differences in behaviour between early-stage (pre-tumorigenic stem cells) and late-stage CSCs/LSCs. The Wnt/ß-catenin pathway is fundamental to preserving the self-renewal ability of early-stage stem cells in leukaemias; breast, lung, and liver cancers; and melanomas, whereas the Notch signalling pathway has been implicated in stemness of late-stage cancer stem cells in AML, breast cancer, colon cancer, and glioblastoma [44,124,125]. Stemness of late-stage CSCs in glioblastoma, colon cancer, and pancreatic cancer involves the Hedgehog signalling pathway [44,126,127].

Haematological malignancies are highly heterogeneous in respect to diversity of clinical presentation, cytogenetics, and molecular profiles, as well as a future outcome that is associated with patient- and leukaemia-related factors [24]. Haematological malignancies arise not only from the genetic alterations in malignant cells, but also due to their communication/symbiotic relationship with the microenvironment. The evolution of the disease is strongly associated with reciprocal communication between stroma and malignant cells, which promotes anti-apoptotic signals in LSCs during their migration to the secondary space [128]. Many studies demonstrated that CSCs are quiescent or slowly dividing, whereas leukaemia progenitors are able to divide rapidly via escaping the dormant state [93]. Indisputably, LSCs hold great importance in the pathogenesis and relapse of leukaemia; thus, haematological malignancies should be treated based on stemness pattern [129]. Furthermore, the heterogeneous LSC population shows diversity at the level of functionality, since there exist sub-colonies that display the unfavourable phenotypes of dormancy, long-term neoplasm propagation, and drug insensitivity. This has modified the understanding of therapeutic needs in haematological malignancies, due to the fact that unfavourable phenotypes of dormancy are reversible and give space to use LSC-targeted treatments that prolong remission periods [130].

Table 4 shows the role of CSCs/LSCs in the recurrence of selected solid and non-solid tumours.

## 4. Perspectives and Modern Therapeutic Strategies Targeting CSCs in Solid Tumours

Despite prominent advances in modern oncology, relevant limitations and challenges still remain. Understanding the unique metabolic properties of CSCs might potentially enhance our ability to manage the therapeutic limitations that CSCs generate. The expected model of CSC-targeted therapies in comparison to conventional therapeutic approaches is presented in Figure 4.

Sekar et al. used liver cancer cell lines—Huh7—and found a reduced expression of CD133 and of the ABCG2 gene in cells treated with XAV939 and silenced with the EpCAM gene. Furthermore, cells treated with cisplatin alone formed spheroids, whereas the EpCAM gene-silenced cells and those treated with XAV939 in combination with cisplatin did not appear as spheroids. In a cytotoxicity assay, cisplatin alone and in combination with EpCAM silencing and XAV939-treated cells showed greater lactate dehydrogenase release than counterparts treated with the XAV939 silenced EpCAM cell group [150]. In their study, Miao et al. used oral squamous cell carcinoma (OSCC) cell lines and multicellular tumour spheroid models to generate CSC-like cells. They performed RNA sequencing to analyse the transcription levels of metabolic genes and analysed the single-cell transcriptome of six OSCC tumours to investigate the metabolic phenotypes of oral CSCs in their native microenvironment in humans. They concluded that CSCs were metabolically inactive compared to differentiated cancer cells and may be resistant to current metabolic therapeutic strategies [151].

In a different work, Huang et al. studied the antiproliferative effect of shikonin in a subpopulation of chemoresistant non-small cell lung cancer. They used A549 sublines to show shikonin’s antiproliferative properties. Shikonin also downregulated the PI3K/Akt/mTOR signalling pathway, inducing apoptosis. They discovered a synergistic action of modest dosages of shikonin and the dual inhibitor BEZ235, which suppressed the growth of lung CSCs and decreased the likelihood of lung cancer recurrence [152]. Furthermore, Santos et al. focused on the mechanism by which the ruthenium–xanthoxylin complex (RXC) targets the Hsp90 chaperone and eradicates colorectal cancer (CRC) stem cells. They demonstrated that RXC is very cytotoxic, inducing apoptosis in primary cancer cells as well as cancer cell lines [153]. In HCT116 CRC cells, Silva et al. investigated the mechanism of action of the ruthenium–5-fluorouracil (Ru/5-FU) complex. Ru/5-FU decreased colonosphere development, the percentage of CD133^+^ cells, and clonogenic survival, suggesting that Ru/5-FU can suppress stem cells in HCT116 cells. Additionally, in vivo HCT116 cell proliferation and experimental lung metastasis in mouse xenograft models were suppressed by Ru/5-FU. The complex inhibits Akt/mTOR signalling, making it a promising anti-CRC chemotherapeutic candidate [154]. Shang et al.’s research in CRC focused on tumour-associated macrophages (TAMs), specifically how they create niches for CSCs. The authors noted that poor treatment outcomes in CRC patients are associated with high expression of inhibitor of differentiation 1 (ID1) in TAMs. They showed that reducing ID1 expression increases the sensitivity of CRC to chemotherapy and immunotherapy [155]. Li et al. claimed that standard anticancer treatment is less effective against CSCs and can even enhance stemness gene expression. They discovered BBI608, a naphthofurandione, which is able to reduce metastasis and disease recurrence, via limitation of spherogenesis and Stat3-driven transcription. There was strong evidence that BBI608 reduced liver metastasis in a xenografted human CRC model and it robustly prevented recurrence in pancreatic cancer. Thus, an unconventional approach increases the range of treatment options for oncology patients, and the procedures based on cancer stemness inhibition open new possibilities for more effective treatment [10].

Zavareh et al. analysed the potential of the endemic plant Satureja bachtiarica in inhibiting and attacking CSCs in glioblastoma and breast cancer. They showed, especially in breast cancer, that S. bachtiarica can be an effective drug that reduces the viability and growth rate of cells, by inducing apoptosis, and it inhibits their migration [156]. Focusing on breast CSCs, Gil-Gas et al. investigated the role of the pigment epithelial-derived factor (PEDF) signalling. They designed a protein that blocks endogenous PEDF in cell culture tests and the modified PEDF interfered with CSC self-renewal and reduced the percentage of CSCs [157].

The aggressiveness of pancreatic cancer is believed to be closely related to a subpopulation of CSCs that have a greater evolutionary ability to escape the cytotoxic effects of chemotherapy compared to other cancer cells. In their work, Mouti et al. demonstrated that using the KMT2A-WDR5 inhibitor to target the protein subcomplex in pancreatic CSCs reduced the cells’ ability to self-renew, their survival, and their ability to cause tumours in vivo [158]. Interestingly, a recent study by Boudreault et al. analysed the role of the TGF-β signalling pathway in melanoma. They showed that TGF-β acts as a potent suppressor of tumour development, migration, and metastasis. Additionally, it has been shown that there is potential in the use of agents that stimulate or mimic TGF-β as new methods to fight melanoma [159].

The concept of CSC-targeting drugs must be taken into account for the reduction in adverse effects and dose-limiting toxicities. Also, for an efficacious therapy, all CSCs should be precisely eradicated to minimise risk of recurrence. In the past few years, a global effort has been made to design innovative therapeutic strategies against CSCs [10,150,151,152,153,154,155,156,157,158,159]. The results seem to be promising to improve long-term health outcomes; however, the biology of CSCs and their susceptibility to various types of therapy depend on the model on which the research was carried out. Additionally, the ability of CSCs to enter a dormant state, as well as the intra-tumoral diversity in surface markers expressed, makes it difficult to attain fully effective solutions. Last but not least, it is difficult to reproduce the real conditions of cancer development in experimental models, which would reflect the complexity of all components relevant for effective anticancer therapy [160]. Therefore, considering all these issues, further research will be necessary in this regard. In addition to discussing the essence of targeted therapies against CSCs in solid tumours, it is also necessary to emphasise the complex issue of anticancer agents that target LSCs in haematological malignancies.

## 5. Agents That Target Leukaemia Stem Cells in Haematological Malignancies

Conventional chemotherapy and stem cell transplantation have augmented the survival of patients with AML, multiple myeloma, and other haematological malignancies, but additional therapeutic strategies are needed [161,162,163]. Cancer stem cells are a logical target for novel drugs and the modulation of oncogenic cell signalling, and metabolic alterations in stem cells have attracted special attention [161,164,165,166]. The Wnt, Hedgehog, NF-κB, and Notch signalling routes play critical roles in the differentiation, proliferation, and survival of cancer stem cells [167,168], so various compounds have been developed targeting these pathways specifically. Cellular therapies have also provided good results in treating haematological malignancies, including targeting stem cells, but will not be covered in this article and are reviewed in the specialised literature [161].

### 5.1. Agents Targeting Wnt and Hedgehog Pathways

The Wnt/ß-catenin pathway promotes the expansion of haematopoietic stem cells and is activated in drug-resistant leukaemia-initiating cells, as demonstrated by different authors [161,169,170,171,172]. C-82 (Figure 5) and ICG-001 are β-catenin/CREB-binding protein (CBP) antagonists that block the interaction between the two proteins, downregulating Wnt-activated genes. Similar to β-catenin silencing with siRNA, those compounds restored the sensitivity of chronic myeloid leukaemia (CML) stem/progenitor cells to tyrosine kinase inhibitors [171,172].

The Hedgehog pathway has also been implicated in resistant phenotypes of CML cells [161,169,173,174]. Vismodegib is a drug targeting the Hedgehog pathway approved for cancer therapy. The incubation of CML cells with vismodegib decreased protein levels of relevant markers like MYC and induced autophagy [173]. Furthermore, the simultaneous inhibition of autophagy strongly enhanced the cell viability decrease induced by vismodegib and triggered apoptosis by way of caspase-3 and -9.

Glasdegib, or PF-04449913, is another clinical inhibitor of the Hedgehog pathway, approved for AML. Sadarangani et al. tested PF-04449913, an antagonist of the GLI2 transcriptional activator, smoothened (SMO), in dormant leukaemia stem cells. The treatment reduced the burden of GLI2-expressing leukaemia stem cells, their dormancy (enhancing cycling), and sensitised the cells to tyrosine kinase inhibition [174].

### 5.2. Agents Targeting NF-κB and Notch Pathways

NF-κB signalling is closely connected to cytokine/chemokine production and immune responses, being recognised a key role in cancer initiation, promotion, and progression [175,176]. The NF-κB pathway is stimulated in cancer stem cells [168,177], and one of the earliest pieces of evidence was the higher NF-κB DNA binding in AML samples compared to normal haematopoietic stem cells [175]. Alone or in cooperation with other signalling pathways, NF-κB promotes the expression of a wide variety of downstream targets, including stem factors (NANOG, SOX2, CD44, and others) and microRNAs, like let-7 and microRNA-21, contributing to self-renewal and expansion features of cancer stem cells [168,175,176,177].

Inhibition of NF-κB signalling with BMS-345541 reduced the stemness, self-renewal, and migration capacity of lung cancer stem cells [168]. BMS-345541 is an inhibitor of IκB kinase and reduced the expression of epithelial-to-mesenchymal transition genes and of the antiapoptotic BAX, along with decreasing the sphere-forming capacity of the cells.

The drug selinexor (Figure 5), described as interfering with NF-κB signalling, has been approved for the treatment of relapsed/refractory multiple myeloma [163]. It inhibits the protein exportin 1, the nuclear exporter of tumour suppressor proteins, the glucocorticoid receptor, and oncoprotein mRNAs, suppressing NF-κB activity, among other effects [178]. In spite of some safety concerns, selinexor in combination with dexamethasone resulted in treatment responses in patients with myeloma refractory to standard therapies [163]. Meanwhile, selinexor combinations with other chemotherapeutics showed the ability to inhibit cancer stem cell spheroids in pancreatic ductal adenocarcinoma [179], and interest in inhibitors of exportin 1 for haematological malignancies is growing [180].

Inflammation and NF-κB activity can crosstalk with the Notch pathway in different ways [177,181]. For example, IL-6-induces Notch1 activation and cancer stem cell proliferation by the assembly of γ-secretase at membrane caveolae [182]. Hence, controlling inflammatory and NF-κB signals can beneficially modulate the Notch-mediated stimulation of cancer stem cells. The addition of γ-secretase inhibitors, namely, MK-0752 [183] and RO4929097 [184], to chemo/radiotherapy gave indications of reducing cancer stem cell populations (CD44^+^, CD24^−/low^, ALDH^high^, and CD133^+^ cells), encouraging their assessment in haematological malignancies. Considering the key role of acute and chronic inflammation, the ability of the polyphenols discussed in the next section to regulate NF-κB signalling harbours great potential for controlling cancer stem cells.

### 5.3. Polyphenols

The chemopreventive action of polyphenols is supported by plenty of in vitro data, as well as by animal studies and epidemiological evidence [164,176]. These compounds usually affect multiple targets, modulating different interconnected biochemical processes, so they can put forward robust mechanisms of action against carcinogenesis and stemness-associated pathways [185,186]. The analysis of a group of 21 phenolic compounds and their interaction with cancer stem cell-related genes pointed to a selection of five high therapeutic potential compounds: resveratrol, curcumin, quercetin, epigallocatechin gallate (EGCG), and genistein [187]. Resveratrol is the chief stilbene present in grapes and wine, being one of the most established anticancer polyphenols [176]. Data from different works show that resveratrol and its methylated derivative pterostilbene can target cancer stem cells, regulating central mediators in signalling pathways [164,188]. Among the several mechanisms of action involved, resveratrol was reported to trigger autophagy via inactivation of Wnt/β-catenin pathway and suppresses the growth of cancer stem-like cells by inhibiting the fatty acid synthase [164,166,188].

Curcumin is another top anticancer polyphenol that displayed relevant effects on models of haematological malignancies [164,176]. It prevented the growth of CD34^+^CD38^−/low^ cells isolated from AML patients by promoting the expression of osteopontin [189]. Burkitt lymphoma and AML cells incubated with low microM concentrations of curcumin exhibit a dose-dependent decrease in markers of cancer stem cells, namely, the ratio of ALDH-positive cells, inhibition of colony formation, and downregulation of Notch1, Gli1, and Cyclin D1 [190]. Curcumin showed strong cytotoxicity towards a human leukemic stem cell line (IC50 of 14 microM), and another curcuminoid, bisdemethoxycurcumin, greatly repressed the expression of Wilms’ tumour 1 and CD34 protein, warranting further studies to control leukaemia stem cells [191]. In this line, Nirachonkul et al. presented an alternative formulation of curcumin in nanoparticles targeting CD123 and, tested in the same leukaemia stem cells, it promoted the polyphenol interaction with the cells and induction of apoptosis, without apparent toxicity to peripheral blood mononuclear cells [192].

Quercetin is a prototypical flavonoid with antioxidant actions at low concentrations and is able to modulate diverse cellular processes underlying cancer initiation and progression. Regulation of microRNAs plays an important role in the anticancer activity of quercetin and, in particular, the upregulation of microRNA-200b-3p was implicated in the inhibition of cancer stem cells [164,193]. At high concentration (50 microM), quercetin interfered with the DNA damage response and inhibited the PI3K/AKT pathway in haematopoietic stem and progenitor cells [194]. Indeed, the inhibition of the PI3K/Akt/mTOR pathway was underlined as a key mechanism of quercetin for the elimination of cancer stem cells [164].

Green tea consumption shows beneficial effects, and EGCG is the component responsible for its stronger molecular anti-carcinogenic actions and results in human trials [176]. There is abundant evidence that EGCG can eliminate cancer stem cells of different types, decreasing stemness markers and inhibiting the Wnt/β-catenin pathway and proliferation indices, among other actions [164,195]. EGCG in combination with quercetin induced apoptosis of prostate cancer stem cells, and inhibited cancer stem cell proliferation phenotypes, in association to caspase activation and downregulation of cell survival mediators [195].

Genistein is a soy isoflavone able to protect haematopoietic stem cells from DNA damage [196]. Mechanistic studies with different cell models pointed to suppression of the Hedgehog/Gli1 pathway and/or upregulation of PTEN as key factors accounting for the anticancer activity of genistein [164].

Apigenin is another flavonoid with interesting anticancer activity, including targeting leukaemia stem cells responsible for failure in AML treatments [165,197]. The combination of apigenin with LY294002 (Figure 5) for treatment of CD34^+^CD38^−/low^ leukaemia cells, including leukaemia stem cells, induced apoptosis in these cells associated to caspase activation, mitochondrial dysfunction, and downregulation of Bcl-xL and NF-κB [197]. Remarkably, these effects were not observed in healthy haematopoietic stem cells, suggesting an option for the safe eradication of leukaemia stem cells. Low sub-toxic concentrations of the two drugs were used and the potent synergistic action was rationalised on the basis of the simultaneous inhibition of the PI3K/Akt pathway by LY294002 and of protein kinase casein kinase 2 (CK2) by the flavone. In accordance, similar effects on caspase-3, antiapoptotic Bcl proteins, and NF-κB were reported with lung tumour and osteosarcoma models treated with apigenin or isovitexin (apigenin glucoside) in vivo [165]. These compounds were also shown to reduce stemness markers, namely, CD133, NANOG, MgSOD, and SOX2, in various in vitro and in vivo cancer models, with implication of c-Met signalling inhibition in the mechanism of action [165,186,198]. The increased expression of microRNA-34a was also associated to the stemness inhibition and apoptosis induction by isovitexin in hepatocellular carcinoma spheroids [199].

The inhibition of CK2, important for the maintenance of cancer stem cells, was also directly implicated in the reduction in self-renewal capability of HeLa sphere-forming cells by apigenin, while downregulation of survival and proliferation factors was accounted for by the sensitisation of CD44^+^ prostate cancer stem cells to cisplatin [165].

In overall, curcumin and apigenin (Figure 5) are the polyphenols showing stronger capacity for regulating cancer stem cells in haematological malignancies.

### 5.4. Other Natural Compounds and Derivatives

In addition to polyphenols, other natural compounds have been shown to inhibit the survival or growth of diverse cancer stem cells. An overview of the effects and mechanisms of action of compounds like sulforaphane, indole-3-carbinol, and phenethyl isothiocyanate can be found in the recent Chu et al. review [164].

Nevertheless, studies with models of haematological malignancies or those comparable uncovered some actions of specific relevance. Withaferin A is a steroidal lactone (Figure 5) and was found to induce cell cycle arrest and apoptotic death of multiple myeloma cancer stem cells [200]. Moreover, it was able to repress the growth and spheroid formation of diverse cancer cells [164].

The alkaloid berberine showed anticancer activity in various conditions, possibly by inhibiting HDACs and modulating the expression of stem cell-associated genes [162,201]. More potent than the flavonoids apigenin and wogonin, the naphthoquinone shikonin increased the apoptosis rate and inhibited invasiveness of renal carcinoma stem cells [186]. Shikonin reduced the expression of diverse cancer stem cell markers like ALDH3A1, CD133, EZH2, NANOG, and SOX2. Moreover, the combination of the phytochemical with an immune checkpoint inhibitor revealed a promising treatment strategy by regulating the T cell population [186]. Parthenolide is a sesquiterpene lactone (Figure 5) that prompted robust apoptosis in leukaemia stem cells, but not in normal haematopoietic cells, by a mechanism associated with inhibition of NF-κB and proapoptotic activation of p53 [202].

The modification of natural compounds is often employed to overcome limitations in the bioavailability or to improve the pharmacological potency. Li et al. synthesised derivatives of apigenin and found one—compound 15e—with strong activity (IC50 of 0.49 versus 44 microM of apigenin) against the growth of human renal carcinoma cells [203]. More recently, Fernando et al. reported a fatty acid ester of phloridzin that inhibited spheroid formation by breast cancer cells in vitro [204]. The conjugate could also inhibit the metabolic activity and induce cell death of paclitaxel-resistant variants, while investigation of the effects on stemness markers at low concentrations is warranted [204]. A synthetic analogue of genistein was also demonstrated to attenuate the expression of FoxM1 and other stemness features of gastric, ovarian, and lung cancer cells [164].

A more specific mechanism of action has been attributed to the gossypol enantiomer AT-101 (Figure 5). This compound is a BH3-mimetic pan-Bcl-2 inhibitor, binding to the BH3 motif of Bcl-2, Bcl-xL, and Mcl-1, in a way that inhibits their anti-apoptotic action, activates Bax and can trigger mitochondrial Smac release. In leukaemia stem cells, it inhibited proliferation and activated the intrinsic apoptotic pathway, with apparently low effects on normal CD34^+^ haematopoietic cells [205]. In accordance with the expected mechanism of action, AT-101 caused a decrease in mitochondrial membrane potential, and DNA damage partially dependent on caspase activity. The compound at microM concentration was also effective in ex vivo AML samples, offering a potential alternative therapy of relapsed and refractory conditions associated to cancer stem cells.

The multifunctional action and safety profile are usually regarded as important advantages of natural compounds like polyphenols for preventive and combination protocols [176,186,197,206,207]. Human trials showed that resveratrol and curcumin doses of up to a few g/day are tolerable, and some cases of hepatoxicity of EGCG have been reported only with oral doses equal or above 800 mg/day [176,206,208].

The above-mentioned astudies shown how important CSCs/LSCs are in cancer progression and as it stands a promising therapeutic target. Based on the works published in 2023–2024, we believe that this topic remains an open field for research and a consensus regarding the nature of CSCs/LSCs has still not been reached.

## 6. Top 10 Reasons Why This Manuscript Is Important in the Oncology Field

Provides a historical graphical overview of research on the nature of CSCs/LSCs.Provides a clear, extensive, tabular presentation of the differences between normal stem cells and CSCs.Underlines the role of the tumour microenvironment in maintaining the pro-tumorigenic ability of CSCs/LSCs.Provides an organised summary of the knowledge regarding the functional methods of CSC/LSC detection including the application, benefits, and weaknesses of selected methods.Specifies the usefulness of the new technologies, including artificial intelligence and deep learning, in CSC examination.Provides an overview of the immunophenotypes of CSCs and LSCs in solid tumours and haematological malignancies.Discusses key characteristics of early-stage (pre-neoplastic) and late-stage (pro-metastatic) cancer and leukaemia stem cells.Indicates the importance of CSCs and LSCs in the recurrence of selected solid and non-solid cancers.Provides a concise analysis of the perspectives and modern therapeutic strategies targeting CSCs in solid tumours.Provides a broad analysis of candidate drugs for regulating LSCs in haematological malignancies, taking into account particular Wnt, Hedgehog, NF-κB, and Notch signalling pathways.

## 7. Methodology

To review the role of cancer stem cells in cancer biology, we carried out large-scale electronic searches within the following public databases: PubMed (U.S. National Library of Medicine) and Google Scholar. The following keywords were used alone or in combination: “cancer stem cells nature/biology”, “history of cancer stem cells”, ”epigenetics in cancer stem cells”, ”cancer stem cells immunophenotype in solid tumours and haematological neoplasms”, ”methods of cancer stem cells detection”, ”tumour microenvironment“, “cancer stem cell signalling pathways“, “epithelial-mesenchymal transition“, “artificial intelligence in cancer stem cells detection“, “cancer stem cells in solid and non-solid tumours recurrence”, “agents targeting leukaemia stem cells”, and “therapy targeting CSCs in solid tumours”.

Screening of the articles was made by four independent authors (BRC, RL, KK, and DM-d-S) and all inaccuracies were detected by final check by (BRC). The resulting literature was analysed and included in our review. The papers’ assessment was based on a critical reading. Only the full-text of articles in English was taken into consideration. Data from the current literature, up to January 2024, including clinical trials, prospective and retrospective observational studies, and review articles were reviewed. Most of the incorporated papers were published in the years 2015–2024 (75%). Since this review was based on previously published research, no ethical approval or patient consent was required.

## 8. Conclusions

Indisputably, CSCs/LSCs differ from normal/haematopoietic stem cells morphologically and functionally. CSCs/LSCs are heterogeneous populations and are influenced by the complex tumour microenvironment. Cancer progression and metastasis are strongly connected with CSCs/LSCs nature and biology. CSCs/LSCs constitute robust populations that can reversibly manoeuvre between different phases of the cell cycle, which gives them the ability to arrest pro-apoptotic signals and prolong the survival in quiescent state. Therefore, there is an urgent need for more sophisticated but easy-to-apply techniques to detect CSCs/LSCs, in order to identify patients who are at high risk for recurrent disease. Advances are also awaited in the clinical translation of the synthetic and natural drugs targeting CSCs discussed in this work.

## Figures and Tables

**Figure 4 ijms-25-03903-f004:**
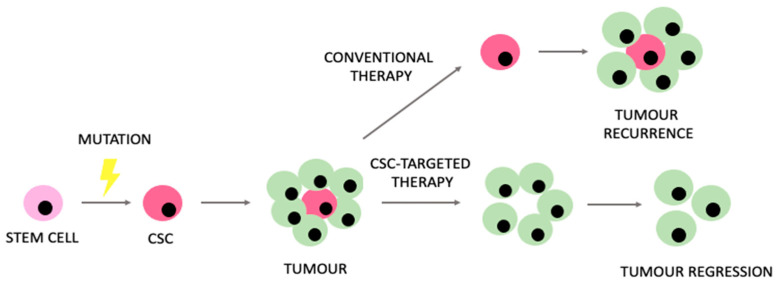
Rationale encouraging the use of therapies targeting cancer stem cells (CSCs) in comparison to conventional therapeutic approaches.

**Figure 5 ijms-25-03903-f005:**
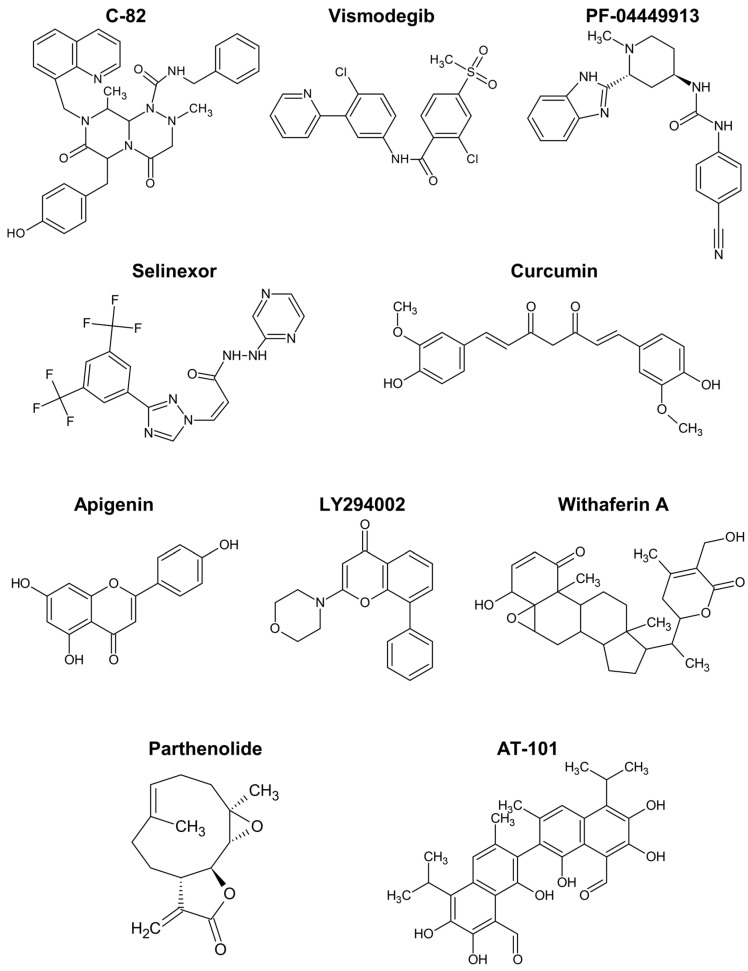
Chemical structure of compounds described to target cancer stem cells in haematological malignancies.

**Table 2 ijms-25-03903-t002:** Summary of complex phenotypes of cancer stem cells (CSCs) and leukaemia stem cells (LSCs) in solid and non-solid tumours.

Cancer	Surface Marker and Cancer-Related Action
Breast cancer (BC)	
	Implantation of only 100 cells with CD44^+^, CD24^−/low^, and lineage immunophenotypes led to breast cancer development [47].
	The CD44^+^ and CD24^−/low^ immunophenotypes were attributed to breast cancer stem cells [48].
	Aggressive triple-negative breast cancer harbours CSCs with the phenotypes CD44^+^, CD24^−/low^, and ALDH1^high^ [49].
	Longer survival rate and lack of lymph node involvement were linked to the phenotype of CSCs CD44^+^ and CD24^−^ [50].
	Detection of ALDH1A3 was related to a pro-metastatic potential [51].
	EpCAM^+^ and CD49f^+^ cell phenotypes in breast CSCs (classified as triple-negative) showed a high tumorigenic ability [52].
Prostate cancer (PC)	
	Overexpression of CD44^+^ marker was associated with uncontrolled proliferation and self-renewal properties [53].
	CD133^+^ and CD44^+^ cells were considered a subfraction of prostate CSCs [54].
	CD44^+^ and CD24^−^ cells demonstrated stem-like properties, including high tumorigenic ability [55].
	ALDH1A1^high^ cells showed high clonogenic and tumorigenic properties; this may serve as a prostate CSC-associated indicator [56].
	CD133^+^ and CD44^+^ cells were able to form spheroids and showed embryo-like attributes [57].
Cervical cancer (CC)	
	CD133^+^, CD44^+^, and ALDH^high^ cells showed high cell division rate and self-renewal properties [58].
	CD49^+^, AII^+^, p63^+^, CK-17^+^, and ALDH^bright^ cells were considered a subfraction of putative cervical CSCs [59].
	CD44^+^ and CD24^−^ cells were considered a subfraction of cervical CSCs [60].
	CD49f^+^, CD71^−^, and CD133^+^ cells were considered a subfraction of cervical CSCs [61].
Ovarian cancer (OC)	
	CD133^+^, CD44^+^, and ALDH^high^ cells showed high cell division rate and self-renewal properties [62].
	CD133^+^ cells demonstrated a high tumorigenic property [63].
	CD44^+^ and CD24^−^ cells were considered a subfraction of ovarian CSCs [64].
	CD133^+^ and ALDH^high^ cells were considered a subfraction of ovarian CSCs [65].
	CD44^+^ and cKIT^+^ cells were attributed a subfraction of ovarian CSCs [66].
Brain cancer (BnC)	
	CD133^+^ cells demonstrated high cell division rate, self-renewal, and pro-angiogenic properties [67].
	Only a few CD133^+^ cells were enough to generate a cancer [68].
	A consensus on the CD133 marker has not been fully established, since tumours can also develop from CD133^−^ cells in gliomas [34].
	CD15^+^, CD44^+,^ CD133^+^, and α6integrinhigh subpopulations demonstrated the highest ability for clonogenic self-renewal in vitro and increased in vivo tumorigenic capacity [69].
Colorectal cancer (CRC)	
	CD133^+^ and CD44^+^ cells were able to form spheroids, migrate, and showed the EMT phenomenon [70].
	The co-expression of CD26^+^, CD44^+^, CD133^+^ was associated with the development of new metastatic tumours [71].
	The co-expression of CD44^+^ and CD133^+^ was associated with synchronous hepatic metastasis [72].
	Colorectal CSCs were confirmed by single or co-expression of CD44^+^ and CD133^+^ surface markers [73].
	CD133^+^ and CD44^+^ cells were able to form spheroids and were resistante to anticancer agents [74].
Lung cancer (LC)	
	Higher CD133 expression was linked to undifferentiated tumours, lymph node involvement, and drug resistance [75].
	ALDH^high^ cells were associated with a higher risk of relapse in locally advanced NSCLC [13].
	ALDH^high^ cells were detected in NSCLC patients and cell lines [76].
Pancreatic cancer (PcC)	
	The phenotypes CD24a^+^, EpCAM^+^, and CD133^+^ of CSCs were associated with a self-supporting model for integrity and maintenance, which promote malignancy [2].
	Higher expression of CD44^+^ and CD133^+^ was associated with a higher risk of relapse and pro-metastatic potential [77].
Liver cancer (LrC)	
	Overexpression of glypican-3, alpha fetoprotein, cytokeratin 19, CD44^+^, CD133^+^, and CD24^+^ were established as liver cancer markers [2,78].
	The phenotypes CD24^−^ and EpCAM^+^ were detected in primary HCC cells as well as primary HCC spheres [79].
	CD133^+^ and EpCAM^+^ cells were able to create viable and dense spheres in comparison to their negative counterparts [80].
Head and neck squamous cell carcinoma (HNSCC)	
	The CD44^+^ surface marker was confirmed in head and neck cancer [81].
	The phenotypes CD44^+^and ALDH^high^ of CSCs were linked to pro-metastatic potential. Additionally, size and advanced stage of primary tumours were associated with a higher number of CSCs [82].
	Higher CD133^+^ expression was linked to higher growth rate, self-renewal ability, and drug resistance [83].
	CD133^+^ and CD44^+^ cells showed high motility, colony formation ability, and potent resistance to anticancer treatment [84].
Acute myeloid leukaemia (AML)	
	CD34^+^ and CD38^−^ cells were able to initiate AML in a mouse model [16].
	The phenotypes of CD34^+^, CD38^−^ cells were considered as leukaemia stem-like cells [85].
	The CD93^+^ marker was indicated on LSCs and is essential for development of MLL-rearranged (current name of the gene KMT2A) AML [86].
	CD34^+^ cell population was considered as functional LSCs [87].
	A higher incidence of recurrence was related to detection of CD34^+^ blasts [88].
	The CD34^+^, CD38^−/low^, and CD123^+^ phenotypes of blasts were associated with worse overall survival [89].
	Cells with the phenotypes CD45^dim^, CD34^+^, CD38^−^, and CD133^+^ were considered LSCs [90].
Chronic myeloid leukaemia (CML)	
	CD25^+^ and IL-1RAP surface markers are specific for LSCs. Both antigens were associated with the activation of NF-kB and AKT signalling pathways, which enhanced proliferation of CML LSCs [91].
	Cells with phenotypes CD45^dim^, CD34^+^, CD38^−/low^, and CD133^+^ were established as leukaemia-initiating cells [90].
	Lin^−^, CD34^+^, CD38^−/low^, CD45RA^−^, cKIT^−^, and CD26+ cells were considered a subfraction of putative CML LSCs [92].
	Lin^−^, CD34^+^, CD38^−/low^, CD90^+^, and CD93^+^ cells were considered a subfraction of chronic-phase CML LSCs [93].
	CD25^+^ was identified as a CML indicator of LSCs and a suppressor of growth [94].
	CD34^+^, CD38^−/low^, and CD26^+^ cells were considered a subfraction of CML LSCs [95].
Acute lymphoblastic leukaemia (ALL)	
	Cells with the phenotypes CD133^+^, CD19^−^, and CD38^−/low^ were considered LSCs [96].
	The percentages of CD34^+^, CD133^+^ or CD34^+^, and CD82^+^ cells in ALL patients were higher than those in healthy volunteers [97].
	Cells with phenotypes CD34^+^, CD38^+^, and CD19^+^, as well as CD34^+^, CD38^−/low^, and CD19^+^ cells, were considered LSCs with self-renewal ability [98].
	In MLL(KMT2A)-AF4 patients, CD34^+^, CD38^+^, and CD19^+^ phenotypes and CD34^−^ and CD19^+^ cells were able to trigger leukaemia, but in MLL(KMT2A)-AF9 patients, CD34^−^ and CD19^+^ cells were considered LSCs [99].
Myelodysplastic syndromes (MDS)	
	The phenotypes CD34^+^, CD38^−/low^, and CD123^+^ confirmed malignant clonal cells with abnormal differentiation, uncontrolled proliferation, and limited apoptosis [100].
	CD34^+^, CD38^−/low^, and CD90^+^ cells demonstrated 5q deletion upon diagnosis and were selectively resistant to treatment [101].
	Higher expressions of Lin^−^, CD34^+^, CD38^−/low^, CD90^+^, and CD45R^−^ cells were shown in cases with the monosomy of chromosome 7 (−7) and deletion of the long arm of chromosome 20 (20q−) [102].
Multiple myeloma (MM)	
	Positive expression of CD24 was considered a dominant marker of MM stem cells, and CD24^+^ cells showed self-renewal and drug resistance properties [103].
	ALDH^high^ cells had upregulated chromosomal instability genes associated with low drug sensitivity and high tumorigenic rate [104].
	Clonotypic CD138^−/low^ cells exhibited robust stemness characteristics, drug resistance, and anti-apoptotic potential and a higher ability to sustain in G0 and G1 cell cycle phases [105,106,107].
	Cells with the phenotypes ALDH^high^ and CD138^−/low^ presented high potential to generate tumours [105,108].

ALDH: aldehyde dehydrogenase, normally represent a higher mitotic index, colony forming capacity, self-renewal, in vivo tumorigenic and dissemination capacity, and low drug sensitivity; EpCAM: epithelial cell adhesion molecule; HCC: hepatocellular carcinoma, NSCLC: non-small cell lung cancer; CD19: molecule, B-Lymphocyte Surface Antigen B4; CD24: a small surface protein responsible for the cell–extracellular matrix (ECM) and cell–cell interactions; CD34: a transmembrane glycoprotein expressed on early lymphohematopoietic stem cells, progenitor cells, and endothelial cells; CD38: a multifunctional transmembrane protein that is a lymphocyte receptor; CD44: a multifunctional glycoprotein responsible for cell adhesion, signalling, proliferation, migration, haematopoiesis, and lymphocyte activation; CD45RA: a specific marker for leukaemia stem cell subpopulations in acute myeloid leukaemia; CD90: a glycophosphatidylinositol (GPI) anchored conserved cell surface protein; CD93: indicates pro-leukemic cells (leukaemia-initiating cells) and stimulates LSC proliferation; CD123: interleukin-3 receptor alpha chain; CD133: also known as prominin-1, a transmembrane cell surface glycoprotein commonly utilised as a hematopoietic stem cell marker; CD138: a cell adhesion molecule and a marker in poorly differentiated B cells.

**Table 3 ijms-25-03903-t003:** Characteristics of CSC/LSC subfractions.

Features	Early-Stage (Pre-Tumorigenic)	Late-Stage (Pro-Metastatic)
Cell cycle regulation	Quiescent	Active
Cell division	Mostly asymmetric	Mostly symmetric
Self-renewal capacity	Potent	Potent
Mutation/chromosomal status	Normal	Abnormal (genetic instability)
Tumorigenic ability	Low	Potent
Clonogenic ability	Low	Potent
Migration ability	Low	Potent
Proangiogenic potential	Low	High
Resistance to anticancer treatment	Intrinsic	Both intrinsic and acquired

**Table 4 ijms-25-03903-t004:** The role of cancer stem cells (CSCs) and leukaemia stem cells (LSCs) in the recurrence of solid and haematological cancers.

Name of Cancer	Sample	Key Findings
Breast cancer (BC)		
	MDA-MB-468 basal breast cancer cells.	CSCs showed notable changes, such as enrichment in transduction cascades linked to apoptosis, cellular growth, proliferation, and stemness. AURKB, INCENP, and BIRC5, among other coregulated chromosomal passenger proteins, were overexpressed in CSCs. Overexpression of BIRC5 boosted the population of CSCs in vitro and in vivo. This coregulated module was shown to be overexpressed in basal breast tumours and was also linked to relapse-free and overall survival in patients, according to analysis of previously reported cohorts [131].
	Tumour samples from patients with ER+ breast cancer.	Breast CSCs are enriched in the arterial niche for human oestrogen receptor and interact with arterial endothelial cells; this interaction is driven by the lysophosphatidic acid/protein kinase D signalling pathway. This pathway promotes both EC arterial differentiation and self-renewal. Targeting the LPA/PKD-1-CD36 signalling pathway may inhibit tumour progression by disrupting the arterial niche and eradicate CSCs effectively [132].
Prostate cancer (PC)		
	LNCaP (CRL-1740), HEK 293T (CRL-11268), PC-3 (CRL-1435), and DU145 (HTB-81) cells.	Intracellular domain of JAG1 (JICD) enhances the androgen independence of androgen receptor signalling in prostate cancer cells and, by promoting PC stem-like cell characteristics, migration, and invasion of PC cells, also promotes carcinogenesis. JICD plays a role in the development of PC cells into advanced metastatic castration-resistant prostate cancers [133].
Cervical cancer (CC)		
	Cohort of 332 patients.	The five-year overall survival (OS) and disease-free survival (DFS) rates were longer in the P16INK4A^high^ expression group compared to the P16INK4A^low^ expression group. Five-year OS and DFS rates were shorter in the P16INK4A^low^, SOX2high and P16INK4A^low^, and ALDH1A1^high^ groups, respectively, than in the P16INK4A^high^, SOX2^low^ and P16INK4A^high^, and ALDH1A1^low^ groups. A promising target for patients with cervical cancer is lower P16INK4A expression, which is linked to greater CSC markers and indicates worse future outcomes [134].
Ovarian cancer (OC)		
	Database with 558 ovarian cancer tumour samples. Data retrieval, clinical and pathological features, data pre-processing.	Higher platinum sensitivity was revealed by the mRNA expressions of ALDH1A1 and LGR5. POU5F1 mRNA expression identified tumours resistant to platinum. Longer OS was correlated with the expression of CD44 and EPCAM mRNA, while reduced OS was linked to the levels of THY1 mRNA and protein. The three factors EPCAM, LGR5, and CD44 have a beneficial impact on DFS. The median overall survival in the high-risk group was 9.1 months longer than in the low-risk group in a multivariate model based on CSC marker expression. The expression of ALDH1A1, CD44, EpCAM, LGR5, POU5F1, and THY1 in OC was proposed to predict treatment response and serve as prognostic markers for future outcomes [135].
	Ovarian cancer cell lines Caov3, Ovcar5, and Ovcar8.	The expression of AhRR and PPP1R3C negatively correlates with the OS of patients with OC and progression-free survival. Increased expression of AhRR and PPP1R3C was maintained in some CSC subpopulations, strengthening their potential role in OC [136].
	Cohort of 45 patients affected by third–fifth relapsed ovarian cancer.	Patients with recurrent OC treated with high cell-killing chemotherapy experienced improvements in median progression-free survival (PFS) corresponding to 5.4 months (third recurrence), 3.6 months (fourth recurrence), and 3.9 months (fifth recurrence). Additionally, they showed that patients who did not respond to treatment (CSC drug response test) had a 30 times greater risk of death compared to treatment responders [137].
Brain cancer (BnC)		
	Human glioblastoma (GBM) samples.	Immunoglobulin G (RW03-IgG), dual antigen T cell engager (DATE), and chimeric CD133-specific antigen receptor T cell (CART133) showed activity against patient-derived CD133^+^ GBM cells. CART133 cells demonstrated superior efficacy in patient-derived GBM xenograft models without causing adverse effects on normal CD133^+^ haematopoietic stem cells in humanised CD34^+^ mice [138].
	Human astrocytomas of WHO grade I–IV.	Among astrocytomas, OCT4, MYC, and KLF4 mRNA expression increased with tumour malignancy, while in recurrent gliomas, MYC expression slightly decreased. Moreover, there was a positive correlation between different stem cell markers. Embryonic markers were detected at similar levels in glioma cell lines (long- and short-term cultures). Increased expression of KLF4 (and lower Nanog and OCT4) was observed after exposure to temozolomide [139].
Colorectal cancer (CRC)	Cohort of 797 patients with stage II and III colorectal cancer.	High SOX2^+^ cell density was not associated with poor overall survival. Furthermore, a significant improvement in survival was observed in all patients after treatment with 5-fluorouracil (FU) (regardless of SOX2^+^ cell density). SOX2 can predict response to oxaliplatin but not 5-FU treatment [140].
Lung cancer (LC)		
	Cohort of 118 patients with non-small cell lung cancer.	In 53.7% of samples positive at the time of primary diagnosis, and 25.6% in the case of recurrence, the most prevalent transcript was EpCAM. EpCAM and CK19, NANOG, PROM1, TERT, CDH5, FAM83A, and PTHLH were associated with worse OS. Only CSC-specific NANOG and PROM1 were associated with outcomes at initial diagnosis and disease progression [141].
	Cohort of 35 patients with non-small cell lung cancer.	CSC rate had no impact on the likelihood of a recurrence. In a secondary study, patients with locally advanced cancer and a greater prevalence of CSCs had a higher chance of disease recurrence; patients with early-stage disease did not show this association [13].
Pancreatic cancer (PnC)		
	Human pancreatic cancer cell line Capan-1, MIA PaCa-2, PANC-1, and BxPC-3 cells.	No significant differences were found in the effect of different concentrations of gemcitabine on CD44^+^ or EpCAM^+^ CSCs of different pancreatic ductal adenocarcinoma (PDAC) cell line cultures (BxPC-3, Capan-1, and PANC-1), nor between CSCs and non-CSCs. The expression of the ABCG2 transport protein was significantly higher in CD44^+^ and EpCAM^+^ CSCs of PDAC cell lines. Additionally, CSCs showed low anticancer drug sensitivity. Gemcitabine-resistant PnC cells were associated with epithelial–mesenchymal transition (EMT), a more aggressive and invasive phenotype of many solid tumours. Increased c-Met phosphorylation may also be associated with chemotherapy and EMT resistance and could be a chemotherapeutic target in PnC [142].
Liver cancer (LrC)		
	TCGA (The Cancer Genome Atlas) liver cancer RNA-seq (LIHC) data.	The expression of approximately 30% of genes involved in the glucose metabolism pathway was found dysregulated, with downregulation in hepatocellular carcinoma. Differentially expressed genes are associated with advanced clinical stage and poor prognosis. Furthermore, clustering analysis of differentially expressed genes revealed a subset of patients with a worse prognosis, including reduced OS, disease-specific survival, and recurrence-free survival. This aggressive subtype significantly increased expression of stemness-related genes and downregulated metabolic genes, also increasing immune infiltration, which contribute to poor prognosis [143].
Head and neck squamous cell carcinoma (HNSCC)		
	Cohort of 58 patients.	Progression-free survival was shorter for patients with CD44 positive expression of CSCs [144].
	Cohort of 85 patients with advanced stage HNSCC.	Patients with high CD44 expression showed worse future outcomes, regardless of the survival model application [145].
	Cohort of 40 patients.	High expression of ALDH1 was associated with lymph node involvement and shorter survival rate. This observation confirms the existence of an elevated number of stem-like cells with invasion ability, which are able to promote lymph node metastasis [146].
Acute myeloid leukaemia (AML)		
	Cohort of 121 patients.	Overall survival was shorter for patients with higher enumeration of leukaemia progenitor population [87].
	Cohort of 250 patients.	In CD34^+^ AML subjects, the percentage of the CD34^+^ and CD38^−/low^ cells at diagnosis was associated with shorter patient survival [85].
	Bone marrow aspirates were analysed from 87 patients and 27 healthy donors.	In AML patients, a higher percentage of CD45^dim^, CD34^+^, CD38^−/low^, and CD133^+^ cells (≥40%) was considered an independent prognostic factor for overall survival. Additionally, the immunophenotypes of CD45^dim^, CD34^+^, CD38^−/low^, and CD133^+^ cells allowed for discrimination between LSCs and normal haematopoietic stem cells, as well as emerging as a promising therapeutic approach in AML [90].
	Bone marrow samples were analysed from 111 AML de novo diagnosed patients.	A high percentage (>1%) of CD34^+^, CD38^−/low^, and CD123^+^ cells was associated with poor disease-free survival, overall survival, and treatment failure, regardless of the patient’s cytogenetic profile [25].
	Bone marrow or peripheral blood samples were analysed from 25 AML patients.	After disease recurrence, 9 to 90 times higher LSC activity was observed, independently of the surface markers applied to specify the LSCs. Recurrence after standard chemotherapy was associated with accumulation of more phenotypically composite LSCs. This observation may explain drug resistance and shorter survival rate in patients who relapse after initial treatment [147].
Chronic myeloid leukaemia (CML)		
	Bone marrow aspirates were analysed from 20 CML patients.	CD34^+^ and CD38^−/low^ stem cells constitute a dominant reservoir of residual BCR-ABL+ cells in patients in remission on imatinib mesylate therapy. The probability of disease relapse is associated with the number of LSCs among the residual BCR-ABL+ cells, re-initiating the leukaemia ability of residual BCR-ABL+ cells, and diversity of bone marrow niches that control leukaemia cell growth [148].
Acute lymphoblastic leukaemia (ALL)		
	In vivo model (mouse bone marrow)	Established an infrequent, long-term quiescent subfraction called label-retaining cells (LRCs) manifesting the unfavourable phenotype of dormancy, in vivo drug insensitivity, and re-initiating leukaemia ability. LRCs are useful as a substitute for recurrence-promoting cells in cases for developing treatment to limit relapse [149].
	Bone marrow aspirates were analysed from 59 ALL patients.	CD82 and CD133 expression at the time of ALL diagnosis was higher in respect to the controls. The hyperdiploid karyotype was associated with upregulation of CD133 mRNA. CD82 and CD133 overexpression was linked to the development of ALL progression. Also, CD133 and CD82 were suggested a therapeutic strategy in paediatric ALL [97].
Myelodysplastic syndrome (MDS)		
	Eight patients with MDS.	5q deletions of CD34^+^, CD38^−/low^, and CD90^+^ cells demonstrated at diagnosis were selectively resistant to treatment at the time of complete clinical and cytogenetic remission and based on follow-up, all patients had recurrence during continued lenalidomide treatment with confirmed clinical and cytogenetic progression [101].
Multiple myeloma (MM)		
	Bone marrow aspirates were analysed from 137 MM patients.	Patients with a high preliminary percentage of CD24^+^ MM cells had more bone lytic lesions and worse progression-free survival and overall survival. Tumorigenic ability of CD24^+^ cells was confirmed in vivo after injection of only 10 cells from MM cell lines. Furthermore, CD24^+^ MM cells exhibited higher expression of iPS/ES genes, including NANOG, OCT4, and SOX2 [103].
	E-cadherin-depleted cells in human MM-derived cell lines RPMI 8226 and NCI-H929.	In MM CSCs, loss of E-cadherin led to either G0/G1 or G2/M blockade, depending on the cellular milieu, by regulating its crucial cell cycle mediators in each phase, and also limited the side population phenotype. A new regulatory system of MM CSCs through the E-cadherin/SOX9 axis could contribute to the long-term cell survival and outgrowth associated with recurrent/refractory MM [106].
	Blood and bone marrow were analysed from 16 MM patients.	CD138^−/low^ cells demonstrated insensitivity to four drugs, including the corticosteroid dexamethasone and the thalidomide analogue lenalidomide. CD138^−/low^ cells presented greater drug efflux ability and vital intracellular drug detoxification efficacy [105].

## Data Availability

Not applicable.

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
