# Peer review of "Cancer Stem Cells from Definition to Detection and Targeted Drugs"

_ijms, 2024, doi:10.3390/ijms25073903_

Round 1

Reviewer 1 Report

Comments and Suggestions for Authors

The authors reviewed cancer stem cells (CSC) and leukemia stem cells (LSC), starting with a historical background and origins of CSC. The clarity in the presentation of tables and figures helps the reader understand and follow the review. The review is both informative and comprehensive. I recommend that the manuscript be published in its current form.

Author Response

The responses to Review no 1

Reviewer No 1: The authors reviewed cancer stem cells (CSC) and leukemia stem cells (LSC), starting with a historical background and origins of CSC. The clarity in the presentation of tables and figures helps the reader understand and follow the review. The review is both informative and comprehensive. I recommend that the manuscript be published in its current form.

Authors: In response to the comments of Reviewer No. 1, we would like to thank you very much for reviewing our manuscript and its positive feedback. We wish you all the best.    

Reviewer 2 Report

Comments and Suggestions for Authors

The review manuscript " Cancer Stem Cells from Definition to Detection and Targeted Drugs" is informative and well written. It may be more interesting if head and neck cancer related CSC information can be involved in this review because this cancer type is also well studied in this field. 

Author Response

We would like to inform you that we have revised our manuscript according to the Reviewers form provided. Below please find attached the list of changes we have introduced.
The responses to Review no 2 

All modifications have been labeled in green.

Reviewer No 2: The review manuscript " Cancer Stem Cells from Definition to Detection and Targeted Drugs" is informative and well written. It may be more interesting if head and neck cancer related CSC information can be involved in this review because this cancer type is also well studied in this field. 

Authors: Thank you for dedicating your time and providing insightful feedback on our manuscript. According to your suggestion we have added studies related to CSCs in head and neck cancer.

Reviewer 3 Report

Comments and Suggestions for Authors

In this review Authors presents 24 studies on cancer/leukaemia stem cells alterations associated with disease recurrence, and they systematize the functional assays, markers and novel methods for CSCs screening. Additionally, Authors emphasizes the CSCs 26 involvement in cancer progression and metastasis. I want to congratulate the authors for review on an interesting and important topic.

My suggestions:

The figures (1,2,3) are difficult to read. Maybe changing the font and colors to brighter ones would improve the quality.

Author Response

Manuscript ID: ijms-2911627 
We would like to inform you that we have revised our manuscript according to the Reviewers form provided. Below please find attached the list of changes we have introduced.
The responses to Review no 3

All modifications have been labeled in yellow.

Reviewer No 3: In this review Authors presents 24 studies on cancer/leukaemia stem cells alterations associated with disease recurrence, and they systematize the functional assays, markers and novel methods for CSCs screening. Additionally, Authors emphasizes the CSCs 26 involvement in cancer progression and metastasis. I want to congratulate the authors for review on an interesting and important topic.
Authors: W odpowiedzi na uwagi Recenzenta nr. 3, bardzo dziękujemy za recenzję naszego manuskryptu, jesteśmy za to bardzo wdzięczni. Życzymy Ci wszystkiego najlepszego.
Reviewer No 3: The figures (1,2,3) are difficult to read. Maybe changing the font and colors to brighter ones would improve the quality.
Authors: According to your suggestion we have modified and improved the quality of figures 1-3.